# Changes in Sugar Sweetened Beverage Intake Are Associated with Changes in Body Composition in Mexican Adolescents: Findings from the ELEMENT Cohort

**DOI:** 10.3390/nu14030719

**Published:** 2022-02-08

**Authors:** Lindsey English, Yanelli R. Carmona, Karen E. Peterson, Erica C. Jansen, Martha María Téllez Rojo, Libni Torres Olascoaga, Alejandra Cantoral

**Affiliations:** 1Department of Nutrition and Dietetics, University of North Dakota, Grand Forks, ND 58202, USA; lindsey.english@und.edu; 2Department of Nutritional Sciences, University of Michigan School of Public Health, Ann Arbor, MI 48109, USA; yanelli@umich.edu (Y.R.C.); karenep@umich.edu (K.E.P.); janerica@umich.edu (E.C.J.); 3Department of Environmental Health Sciences, University of Michigan School of Public Health, Ann Arbor, MI 48109, USA; 4Center for Health and Nutrition Research, National Institute of Public Health in Mexico, Cuernavaca 62100, Mexico; mmtellez@insp.mx (M.M.T.R.); libniavib@gmail.com (L.T.O.); 5Department of Health, Universidad Iberoamericana Ciudad de México, Mexico City 01219, Mexico

**Keywords:** body mass index, waist circumference, body fat percentage, sugar intake, beverages, puberty

## Abstract

Changes in consumption of sugar sweetened beverage (SSBs) have been associated with increased body mass index (BMI), but little work has evaluated the effect on waist circumference (WC) and body fat percentage during adolescence, a period characterized by rapid growth and change in dietary behaviors. We examined the relationship of changes in SSB intake and changes in adiposity over two years in 464 Mexican adolescents. Food frequency questionnaires were used to sum intake of regular soda, coffee with sugar, tea with sugar, sweetened water with fruit, chocolate milk, corn atole, and a sweetened probiotic milk beverage. Linear regression models were used to estimate the associations of changes in SSBs with changes in BMI, body fat percentage, and WC, adjusting for sex, socioeconomic status, screen time, physical activity, age, and change in age. Adolescents who increased their daily SSB intake by >2 serving had a −2.72% higher body fat percentage (95% CI: 0.61, 4.82); a 1–2 serving increase was associated with a 2.49 cm increase (95% CI: 0.21, 4.76) in WC compared with those with no change in intake. Within an adolescent sample, changes in SSB intake were related to concomitant changes in body fat percentage and WC, but not BMI.

## 1. Introduction

In Mexico, sugar-sweetened beverages (SSBs) are the main source of added sugar intake among all age groups and make up a significant portion of total energy intake [1]. Children and adolescents have the greatest intake of SSBs and added sugars in Mexico [1]. In 2012, caloric beverages including unsweetened milk products accounted for 17.5% of total energy intake for children and adolescents ages 1 to 19, with flavored milk beverages, caloric soda, and high-fat milk being the top three contributors [2]. SSB intake among adolescents in Mexico is thought to be high for several reasons, including mistrust of water safety, family members’ preferences for SSBs, low SSB prices, and availability of SSB products compared to unsweetened beverages [3,4,5,6]. Interventions to reduce SSB intake, particularly among adolescents, included a national 10% sugar tax enacted in 2014. While preliminary research suggests reductions since this policy was enacted [7], information on changes in individuals’ SSB consumption is lacking.

Mexico, one of the 38 member countries of the Organization for Economic Cooperation and Development, is second only to the United States in rates of obesity in all age groups, with 32.4% of Mexican adults aged 15 to 74 years classified as obese [8]. In 2012, 36% of Mexican female adolescents and 34% of Mexican male adolescents were classified as overweight or obese [9]. These rates were a considerable increase from 1988, when obesity prevalence among 12- to 19-year-old adolescents was 11.1% [9]. In the Mexican National Health and Nutrition Survey (ENSANUT 2018), the prevalence of overweightness and obesity among adolescents aged 12 to 19 years increased to 38.4% [10]. As rates of overweightness and obesity continue to rise in Mexico, especially among adolescents, there is a greater interest in understanding factors driving incidence and prevalence and how dietary influences such as SSB consumption may contribute to changes in weight status.

Prior evidence within the Mexican population shows a clear link between SSB consumption and adiposity. In Mexican children, Cantoral et al. found that cumulative SSB consumption from 1 to 4 years of age was related to a nearly three-fold increase in risk of general and abdominal obesity at 8 to 14 years for those in the highest tertile of SSB intake (mean ± SD; 253.8 ± 301.1 mL/day) compared to those with the lowest intake (mean ± SD; 121.5 ± 151.6 mL/day) [11]. Among middle-aged Mexican women, another longitudinal study documented a relationship between increases in SSB intake and increases in waist circumference [12]. A few studies in European and US adolescents have shown a positive association between increases in SSB intake and increases in adiposity, including BMI [13,14,15]. Nevertheless, studies that use repeated measures to examine changes in SSB consumption in youth are lacking and prior research largely has examined associations with BMI, which provides an incomplete picture of rapid changes in fat mass and distribution characteristic of adolescence [16,17].

Thus, this study aimed to examine the relationship between changes in SSB intake and changes in body composition in Mexico City adolescents over a two-year period since the national soda tax was enacted in 2014.

## 2. Materials and Methods

### 2.1. Study Population

This study included children from the Early Life Exposure in Mexico to Environmental Toxicants (ELEMENT) cohort, which has been described previously [18]. Pregnant women were recruited from 1997 to 2003 during their first trimester of pregnancy. The 1079 children born to these women were followed from birth to age 5 to study the effects of toxicants, primarily lead, on growth and neurodevelopment. Between 2015 and 2018, we followed-up children, now adolescents, from a sample known as ELEMENT 2011 through late adolescence and included additional similarly aged participants from the original cohorts. Anthropometric, dietary, sociodemographic, and lifestyle behavior data as well as biological samples were collected in 2015 and 2017. Of the 554 adolescents, 464 (83.8%) had complete information on socio-demographic characteristics, dietary intake, physical activity, anthropometry and pubertal development. This study was evaluated and approved by the Research, Ethics and Biosafety Committees of the National Institute of Public Health of Mexico and the Institutional Review Board of the University of Michigan. The research team collected written informed consent and assent from mothers upon their enrollment and from adolescents.

### 2.2. Dietary Assessment

Dietary intake was assessed using a Food Frequency Questionnaire (FFQ) adapted from the 2006 Mexican Health and Nutrition Survey (INSP, 2006) which queried consumption of standard portions (i.e., cups, slice, piece, etc.) of 109 food items by days per week (Never to 7 days). If the participant noted consumption every day, a separate inquiry into the number of times per day (1 to ≥6) was conducted. The FFQ was administered by trained professionals at both the 2015 and 2017 fieldwork visits. This FFQ was used to assess the total intake of SSBs. SSBs variables for total intake in 2015 and 2017 were created by adding the daily intake (mL/day) of corn atole, flavored water and drinks with sugar, chocolate milk, soda, tea and coffee with sugar, and sugar-sweetened probiotic milk beverages (e.g., yakult). For the statistical analysis, change in SSB consumption was calculated as the intake at the 2017 visit minus the intake in 2015.

### 2.3. Anthropometric Measurements

Anthropometry was measured by trained and standardized research staff at both the 2015 and 2017 fieldwork visits. We obtained three measures of adiposity: BMI, body fat percentage, and waist circumference. Weight was measured in kilograms to the nearest 0.1 kg using a digital scale (InBody 270). Height in centimeters was measured to the nearest 0.1 cm using a calibrated stadiometer (Perspective Enterprises). BMI was calculated from the height and weight using the formula kg/m^2^. Body fat percentage was assessed using bioelectrical impedance equipment (InBody 270). A non-stretchable measuring tape (SECA 201) was used to measure waist circumference to the nearest 0.1 cm with two measurements taken and averaged.

### 2.4. Covariates

Covariates were selected as potential confounders based on previous literature [19,20,21]. Years of education and child sex were reported at the time the mother enrolled in the ELEMENT cohort. At the fieldwork visits in 2015 and 2017, the adolescents completed a validated, interviewer-administered questionnaire querying usual frequencies of engaging in 14 moderate (3.0–5.9 metabolic equivalents (METs)) and vigorous (METs ≥ 6) physical activities in the past month, for example, dancing, skating, and volleyball, as well as total weekly screen time use including television, videos, and computer use [22]. Total daily physical activity was quantified by converting usual frequency to minutes per day and summing the minutes of moderate and vigorous physical activities across the 14 items. Pubertal status was assessed using Tanner Stages by trained physicians [23]. For the analysis, we utilized the pubic hair growth stages for both boys and girls evaluated at baseline (2015). As the length of follow-up between the two visits varied from 1.18 to 3.47 years, change in age was considered as a confounder.

Household socioeconomic status was assessed using the 8 × 7 Rule established by the Asociación Mexicana de Agencias de Investigación de Mercados y Opinión Pública (AMAI) [24]. This method comprises eight questions and classifies households into one of seven categories based on the total point values obtained from the questions. The categories from highest to lowest level are A/B, C+, C, C-, D+, D, and E. For the purposes of this study, A/B and C+ were classified as higher SES (*n* = 92); C, C-, and D+ were classified as middle SES (*n* = 146); and D and E were classified as lower SES (*n* = 226).

### 2.5. Statistical Analysis

Variables for the change in body composition between 2015 and 2017 were computed for BMI (kg/m^2^), waist circumference (cm), and body fat percentage by subtracting the measurement in 2017 from the information in 2015.

SSB intake in 2015 and change in body composition were examined in relation to demographic variables, physical activity, and screen time using Wilcoxon or Kruskal–Wallis tests. For these bivariate analyses, the age of the participants was categorized as 9 to <13 years, 13 to <16 years, and 16 to 20 years. Moderate and vigorous physical activity (min/day) and screen time (hrs/week) were also categorized in quartiles.

Multivariable linear regression was used to model the association between the servings of SSBs intake change and the change in body composition in adolescence. Model 1 included no confounders; Model 2 adjusted for age, sex, and SES; and Model 3 was additionally adjusted for continuous physical activity, continuous screen time, and the age change between the two visits. SES but not maternal education was included in Model 2 and Model 3 because a significant overlap between these two variables was observed. Total energy intake was seen as a mediator for SSB intake and diet quality association, and therefore was not considered as a confounder in this analysis.

We identified aberrant and influential values through the evaluation of the studentized residuals and Cook’s distance (*n* = 5); however, removal of these observations did not change our results and they were retained in the final models.

Sensitivity analyses to explore effect modification by pubertal stage were conducted given adiposity changes characteristic of adolescence. Multiple linear regression models relating changes in SSB intake and body composition changes were stratified by early puberty (I–II), mid puberty (III–IV), and post-puberty (V), adjusting for Model 3 covariates.

All analyses were performed using STATA SE version 16.1.

## 3. Results

The characteristics of the population and their baseline intake of SSBs in 2015 are shown in Table 1. Half of the participants were female (53%), and most were 10 years or older (62.5%). The median daily SSB intake in 2015 among all individuals was 303 mL (1.25 cups) with an IQR of 462 mL. In this population, males consumed more SSBs than females (342.9 mL vs. 268.6 mL) and older adolescents consumed more than those in early adolescence (≥16 vs. < 16 years: 320 mL vs. 265.7 to 322.9 mL, respectively). Adolescents with a lower socioeconomic status and a lower maternal education consumed the most. Those reporting higher weekly screen time consumed more SSBs (Q1: 257.1 mL vs. Q4: 342.9 mL), while physical activity was not associated with SSB consumption.

Changes in body composition measures in relation to baseline demographic characteristics are shown in Table 2. Females had a greater positive change over the two-year period for all measures compared to males. Participants who were older at baseline had higher body fat percent change while younger participants had a greater BMI change. No associations were observed with waist circumference.

Changes in SSBs intake from 2015 to 2017 were divided into six categories for the analysis based on changes of one-cup intervals. Category 1 (*n* = 53) included those who decreased their SSB intake by two cups or more (≥−480 mL). Category 2 (*n* = 77) had a decrease of one to less than two cups (≥−240 to <−480 mL). Category 3 (*n* = 124) had a less than one cup decrease (>−240 to <0 mL). Category 4 (*n* = 145) included those who had no change or less than one cup increase in intake (0 to <240 mL); this group was used as the reference. Category 5 (*n* = 37) had an increase of one cup to less than two cups (≥240 to <480 mL). Category 6 (*n* = 28) included those who increased their SSB intake by two cups or more over the two-year period (≥480 mL).

Most participants decreased or had no change in their intake of SSBs from 2015 to 2017. A total of 55% of the participants decreased their intake and another 31% had either no change in their intake or an increase of less than one cup per day. Only 14% participants increased their intake by at least one cup per day.

Table 3 shows the results from the linear regression models with the categories of SSB intake change as the predictors and the changes in body composition indicators as the outcome. There was a positive association between change in SSB intake and body fat percentage change. To illustrate, in the crude model, those who decreased their SSB intake by two or more servings/day over the two-year follow-up period experienced a −1.63% lower body fat percentage change (95% CI −3.17 to −0.07) compared to those who did not change their intake. In comparison, those who increased their intake by two or more cups gained an average of 1.19% more body fat (95% CI −0.80 to 3.17; *p*-trend = 0.002). Similar trends were observed in the adjusted models with statistically significant p-trends (Model 2: *p* = 0.004; Model 3: *p* = 0.008).

There was a similar positive, although non-linear, association between change in SSB intake and waist circumference change. In the crude regression models, those who decreased their SSB intake by two or more servings/day over the two-year follow-up period had a 1.81 cm decrease (95% CI −3.53 to −0.07) in their waist circumference compared to those who did not change their intake. Those who increased their intake by one to two cups increased their waist circumference by 1.17 cm (95% CI −0.81 to 3.15), whereas there was no association in the group who increased their intake by two cups or more. Similar trends observed in the adjusted models with statistically significant *p*-trends (Model 2: *p* = 0.012; Model 3: *p* = 0.023).

Associations of SSB change with BMI change over two years, shown in Table 3, were in the same direction as body fat percentage and waist circumference, but did not reach statistical significance after accounting for potential confounders in the adjusted models.

Sensitivity analyses shown in Table 4 considered potential effect modification by pubertal status on the association of two-year changes in SSB intake and body composition. SSB changes were related to changes in body fat percentage and waist circumference during early puberty and late puberty but not mid-puberty. For example, among early-puberty adolescents who increased their SSB intake by at least 480 mL, body fat percentage increased by 4.9% compared with those whose SSB intake did not change over two years.

## 4. Discussion

Within this cohort of Mexican adolescents, the majority (55%) of the sample decreased their SSB intake over a two-year period. Compared to those whose consumption levels did not change, adolescents who decreased their intake had concomitant decreases in waist circumference and body fat percentage, while those who increased their intake also demonstrated increases in waist circumference and body fat percentage. Associations with BMI were in the same direction but not statistically significant. The magnitude of increases in body fat percentage associated with increases in SSB intake was larger among children in early and late stages of puberty as opposed to those in the midst of puberty. This is, to our knowledge, the first study examining changes in SSB intake and changes in BMI, body fat percentage, and waist circumference during adolescence in an exclusively urban Latin American population.

Our findings are in line with some previous literature on changes in SSB intake and changes in adolescent body fatness and distribution. In a randomized control trial (RCT) examining the effectiveness of an intervention among adolescents to reduce SSB intake on the change in BMI and body fat percentage, Ebbeling et al. found that intervention participants reduced SSB intake by 1.3 servings per day compared to those in the control group (who received no education or alternative beverage choices) who had a 0.9 servings per day reduction [25]. In post hoc analyses, these authors found greater reductions in BMI for Hispanic participants (−2.35 kg/m^2^, *p* = 0.01) but not for non-Hispanic participants (0.18 kg/m^2^, *p* = 0.68). No significant findings were seen in body fat percentage [25]. Other RCTs have shown that even short-term increases in sucrose-sweetened beverage consumption can lead to increases in weight and body fatness [26,27]. These trials were conducted in previously overweight adults, however, and provided a substantial increase in caloric intake via sucrose supplements, rather than modest changes in SSB consumption that may be more reflective of free-living conditions.

Several observational studies have found a similar relationship between SSB intake and adiposity. In the ELEMENT cohort, we previously showed that cumulative SSB intake from one to four years of age was associated with a nearly three-fold risk of central and abdominal obesity at 8–11 years [11]. Among middle school children, Ludwig et al. found that for each additional serving of SSB intake, BMI change increased by 0.24 kg/m^2^ over two years [14], and Laverty et al. documented associations of increased SSB intake with increased BMI and increased body fat percentage change over a four-year period [28]. Other studies focusing on adult populations have found similar results with decreases and increases in SSB intake relating to corresponding changes in BMI, body fat percentage, and waist circumference [2,29,30].

We found no statistically significant associations between SSB change and BMI change, consistent with some previous reports. Vaneslow et al. found no association of BMI change over a five-year period for adolescents who increased their soft drink intake by at least one cup per day [31], and Laurson et al. observed similar findings when looking at baseline SSB intake and changes in BMI over an 18-month period [32]. Body fat percentage and waist circumference were more affected than BMI in Mexican adolescents in the present study, underscoring implications of SSB consumption, since android obesity is associated with negative cardiometabolic outcomes [17,33,34].

Several mechanisms may explain a link between increases in SSB intake and increases in adiposity. Increasing SSB intake without decreasing intake of overall calories can contribute to a positive energy balance. Intake of calories from liquids is also believed to have lower satiety compared to whole foods [35]. SSBs lack fiber and protein, nutrients known to be satiating [36]. Sugar-sweetened beverages also have a high glycemic index (GI) that can cause sharp drops in blood glucose, resulting in cravings and excess intake of high and quick energy foods such as simple carbohydrates [37,38]. Long-term excess intake of high-fructose foods such as soft drinks is also linked to increased risk of insulin resistance, impaired metabolism, and weight and fat gain [39,40]. However, the relationship between insulin resistance and body weight and fat gain appears to be interdependent and needs further investigation [41]. Finally, genetics and ethnic background may play a role in both SSB consumption and obesity. Genetic variants more common in Hispanics compared to other ethnic groups link excess SSB consumption to a greater risk of obesity and negative cardiometabolic health outcomes [42,43].

The intake of SSBs within our population is consonant with consumption in urban Mexican adolescents. According to the most recent National Health and Nutrition Survey in Mexico (ENSANUT 2018), 85% of adolescents in Mexico City reported regular consumption of non-dairy SSBs and 6.1% reported consumption of dairy-containing SSBs [10]. At both time points, >90% of our participants reported regular consumption of any SSBs. Yet, we found that on average, the adolescents in the cohort decreased overall SSB intake by about 70 mL over the two-year follow-up period.

The decreases in SSB intake observed in our population could be due to several factors. Adolescence is a transformative period where many start making decisions by themselves, especially regarding food choices. Leal et al. found changes in dietary patterns over a five-year period from childhood to adolescence in their population of Brazilian schoolchildren [44]. Factor scores of an “unhealthy” diet pattern (high in salty snacks, SSBs, fast food, etc.) decreased over the time period while other diet patterns remained unchanged [44]. Another possible explanation is the influence of the sugar tax. In Mexico, SSB purchases among adults and adolescents decreased over 10% in 2014 after the tax was enacted [45]. Considering adolescents have the greatest intake of SSBs in Mexico, the tax may have a greater impact on this population. Reductions in purchases and therefore intake of SSBs in the general population could lead to an overall reduction in Mexico’s adolescent obesity prevalence. Mexico also introduced warning labels on “unhealthy” food items in another effort to deter purchases of high-calorie, high-sugar foods. However, this policy was only recently adopted in late 2019 and little research on its effectiveness exists thus far.

In addition, bottled water purchases increased by 11.5% among this age group [45]. Ng et al. found similar results in their study, finding purchases of high-calorie, high-sugar beverages decreased by 7% in 2014 and almost 19% in 2015 [7]. The reduction of purchases provides more evidence of the impact of SSB intake change on adiposity changes as theoretical models of the tax’s impact on population weight loss show a 2.54% reduction in obesity prevalence over a 10-year period from 2014 to 2024 [46]. Although both of our study visits occurred after the implementation of the tax, it is possible that declines in SSB purchases and intake are gradual and ongoing.

One of the strengths of this study is its longitudinal design, which provides stronger causal inference by observing concurrent changes in the exposure and outcome. Nonetheless, we cannot discount the possibility that changes in adiposity preceded changes in SSB consumption. Another strength is the inclusion of all sugar-sweetened beverages in total SSB intake, not just intake of soft drinks. Although regular soda was the most common SSB consumed, accounting for other popular SSB drinks gives greater context to the effect of SSBs on weight and adiposity. It is especially important to account for drinks common in Hispanic cultures, including agua fresca (water sweetened with fruit and sugar) and atole (corn-based sweetened beverage), as those items contributed to the average total SSB intake both in 2015 and 2017.

There are a number of limitations to this study. The first is the use of a self-reported FFQ, which relies on participant recall and is subject to self-reported bias. However, the FFQ used was validated for use in an adolescent population and is the most common way to collect information on SSB consumption. Sensitivity analyses suggested effect modification of associations of SSB changes with body composition by pubertal progression, but the sample size limited power to conduct analyses stratified by pubertal stage and sex. Generalizability may be limited to adolescents living in urban Latin American settings.

## 5. Conclusions

Overall, this study showed that adolescents who increased their SSB intake also had increases in body fat and waist circumference compared to those with no change in SSB consumption. These findings highlight the period of adolescence as an important stage when dietary changes may result in concomitant changes in body fat distribution. Further investigations are required to understand whether overall declines in SSB consumption and concomitant decreases in android obesity among adolescents in Mexico are a national and ongoing trend.

## Figures and Tables

**Table 1 nutrients-14-00719-t001:** Baseline characteristics of study participants and SSB intake in 2015 (*n* = 464).

	*n* (%)	Median, IQR	*p*-Value
**Sex**			**0.008**
Male	219 (47.20)	342.86, 462.86	
Female	245 (52.80)	268.57, 377.14	
**Age (years)**			**0.024**
9 to <13	153 (32.97)	265.71, 365.71	
13 to <16	156 (33.62)	322.86, 461.43	
≥16	155 (33.41)	320.00, 497.14	
**Maternal Education (years** **)**			
≤9	174 (37.50)	342.86, 400.00	**0.0001**
10 to <12	59 (12.72)	474.29, 634.29	
12	159 (34.27)	291.43, 437.14	
>12	72 (15.52)	171.43, 268.57	
**Socioeconomic Status**			**0.021**
Higher	92 (19.83)	240.00, 442.86	
Medium	146 (31.47)	282.86, 462.86	
Lower	226 (48.71)	340.00, 420.00	
**Physical Activity (min/day)**			
Q1, 15.48 to 53.24	116 (25.00)	322.86, 434.29	0.723
Q2, 53.6 to 69.51	116 (25.00)	291.43, 437.14	
Q3, 70.14 to 90.57	116 (25.00)	305.71, 514.29	
Q4, 90.65 to 216.3	116 (25.00)	297.14, 397.14	
**Screen Time (h/wk)**			
Q1, 1 to 23	119 (25.65)	257.14, 351.43	**0.004**
Q2, >23 to 34	119 (25.65)	302.86, 462.86	
Q3, >34 to 48	110 (23.71)	320.00, 565.71	
Q4, >48 to 108.5	116 (25.00)	342.86, 485.71	

**Table 2 nutrients-14-00719-t002:** Body composition changes from 2015 to 2017 by baseline characteristics (*n* = 464).

	BMI Change		Body Fat Percent Change		Waist Circumference Change (cm)	
	Mean ± SD	*p*-Value	Mean ± SD	*p*-Value	Mean ± SD	*p*-Value
**Sex**						
Male	**1.05 ± 1.86**	**0.035 ***	**−0.96 ± 5.37**	**<0.001 ***	**4.86 ± 5.59**	**<0.001 ***
Female	**1.4 ± 1.66**		**2.34 ± 3.88**		**7.29 ± 5.2**	
**Baseline Age (years)**		**0.021 ***		**0.011 ***		0.094
9 to <13	**1.55 ± 1.66**		**0.17 ± 5.76**		6.83 ± 5.32	
13 to <16	**1.01 ± 1.85**		**0.42 ± 4.8**		5.47 ± 5.59	
>16			**1.74 ± 3.95**		6.13 ± 5.59	
**Maternal Education (years)**		0.752		0.426		0.396
9 or less	1.35 ± 1.79		1.18 ± 4.74		6.65 ± 5.66	
10 to <12	1.17 ± 1.83		0.09 ± 5.4		5.99 ± 5.35	
12	1.14 ± 1.71		0.53 ± 4.63		5.62 ± 5.36	
>12	1.23 ± 1.78		0.92 ± 5.56		6.18 ± 5.64	
**Socioeconomic Status**		0.355		0.203		0.144
Higher	1.21 ± 1.61		1.07 ± 4.7		6.24 ± 5.19	
Medium	1.08 ± 1.82		0.18 ± 5.26		5.42 ± 5.69	
Lower	1.35 ± 1.78		1.05 ± 4.78		6.57 ± 5.5	
**Physical Activity (min/day)**		0.128		0.450		0.339
Q1, 15.48 to 53.24	1.2 ± 1.65		1.17 ± 4.5		5.7 ± 5.56	
Q2, 53.6 to 69.51	1.15 ± 1.69		1.08 ± 4.43		6.26 ± 5.3	
Q3, 70.14 to 90.57	1.04 ± 1.82		0.61 ± 4.95		5.74 ± 5.79	
Q4, 90.65 to 216.3	1.56 ± 1.86		0.25 ± 5.73		6.86 ± 5.39	
**Screen Time (hrs/wk)**		0.592		0.550		0.402
Q1, 1 to 23	1.33 ± 1.41		1.1 ± 4.55		6.67 ± 5.1	
Q2, >23 to 34	1.36 ± 1.92		1.04 ± 5.35		6.45 ± 5.78	
Q3, >34 to 48	1.11 ± 1.87		0.67 ± 5.21		5.74 ± 5.48	
Q4, >48 to 108.5	1.14 ± 1.83		0.27 ± 4.57		5.66 ± 5.68	

Notes: * *p* < 0.05. Two-sample *t* tests were used to compare the body composition changes by sex, and one-way ANOVA tests were used for the age, maternal education, physical activity, and screen time comparisons.

**Table 3 nutrients-14-00719-t003:** Linear regression models between beverage consumption change and body composition changes from 2015 to 2017 (*n* = 464).

	Model 1	Model 2	Model 3
	β (95% CI)	β (95% CI)	β (95% CI)
**Body Mass Index Change**			
Decrease ≥480 mL	−0.44 (−0.72, 0.25)	0.16 (−0.45, 0.78)	0.14 (−0.47, 0.76)
Decrease 240 to <480 mL	−0.24 (−0.72, 0.25)	0.22 (−0.35, 0.78)	0.17 (−0.4, 0.74)
Decrease <240 mL	−0.17 (−0.59, 0.25)	0.36	0.29 (−0.28, 0.85)
No change to increase <240 mL	Reference	Reference	Reference
Increase 240 to <480 mL	0.14 (−0.49, 0.78)	0.55 (−0.19, 1.29)	0.47 (−0.27, 1.22)
Increase ≥480 mL	0.03 (−0.68, 0.74)	0.45 (−0.36, 1.25)	0.42 (−0.39, 1.22)
*p*-trend	0.063	0.089	0.139
**Body Fat Percent Change**			
Decrease ≥480 mL	**−1.63 * (−3.17, −0.07)**	0.85 (−0.76, 2.45)	0.73 (−0.88, 2.33)
Decrease 240 to <480 mL	−0.79 (−2.15, 0.56)	1.01 (−0.47, 2.49)	0.77 (−0.73, 2.26)
Decrease <240 mL	−0.44 (−1.62, 0.73)	1.39 (−0.06, 2.85)	1.10 (−0.37, 2.58)
No change to increase <240 mL	Reference	Reference	Reference
Increase 240 to <480 mL	0.69 (−1.08, 2.46)	**2.01 * (0.06, 3.94)**	1.76 (−0.18, 3.7)
Increase ≥480 mL	1.19 (−0.8, 3.17)	**2.71 * (0.59, 4.82)**	**2.72 * (0.61, 4.82)**
*p*-trend	**0.002 ***	**0.004 ***	**0.008 ***
**Waist Circumference Change**			
Decrease ≥480 mL	**−1.81 * (−3.53, −0.07)**	0.19 (−1.69, 2.07)	0.12 (−1.76, 2.00)
Decrease 240 to <480 mL	−1.46 (−2.98, 0.05)	0.98 (−0.76, 2.71)	0.84 (−0.9, 2.59)
Decrease <240 mL	−0.54 (−1.85, 0.78)	1.39 (−0.31, 3.10)	1.18 (−0.55, 2.91)
No change to increase <240 mL	Reference	Reference	Reference
Increase 240 to <480 mL	1.17 (−0.81, 3.15)	**2.72 * (0.45, 4.98)**	**2.49 * (0.21, 4.76)**
Increase ≥480 mL	−0.24 (−2.46, 1.98)	1.43 (−1.04, 3.89)	1.34 (−1.12, 3.80)
*p*-trend	**0.006 ***	**0.012 ***	**0.023 ***

Notes: * *p* < 0.05. Two-sample *t* tests were used to compare the body composition changes by sex, and one-way ANOVA tests were used for the age, maternal education, physical activity, and screen time comparisons.

**Table 4 nutrients-14-00719-t004:** Linear regression between body composition change and SSB intake change, stratified by pubertal stage (*n* = 464).

	Early Puberty (*n* = 148)	Mid Puberty (*n* = 223)	Post Puberty (*n* = 93)
	β (95% CI)	β (95% CI)	β (95% CI)
**Body Mass Index Change**			
Decrease ≥480 mL	0.51 (−0.79, 1.8)	−0.36 (−1.25, 0.53)	0.40 (−0.84, 1.63)
Decrease 240 to <480 mL	0.19 (−1.04, 1.41)	−0.02 (−0.82, 0.79)	0.79 (−0.36, 1.95)
Decrease <240 mL	0.70 (−0.48, 1.88)	0.05 (−0.75, 0.86)	0.40 (−0.84, 1.65)
No change to increase <240 mL	Reference	Reference	Reference
Increase 240 to <480 mL	0.71 (−0.86, 2.27)	0.13 (−0.9, 1.15)	1.58 (−0.02, 3.18)
Increase ≥480 mL	1.09 (−0.63, 2.81)	0.31 (−1.04, 1.66)	0.09 (−1.29, 1.48)
*p*-trend	0.146	0.405	0.545
**Body Fat Percent Change**			
Decrease ≥480 mL	2.67 (−0.88, 6.21)	−1.05 (−3.05, 0.95)	1.65 (−0.79, 4.09)
Decrease 240 to <480 mL	2.90 (−0.45, 6.24)	−0.60 (−2.4, 1.21)	2.13 (−0.16, 4.42)
Decrease <240 mL	3.11 (−0.13, 6.34)	0.71 (−1.08, 2.51)	1.59 (−0.87, 4.06)
No change to increase <240 mL	Reference	Reference	Reference
Increase 240 to <480 mL	**4.72 * (0.43, 8.99)**	−0.35 (−2.64, 1.95)	**5.63 * (2.45, 8.80)**
Increase ≥480 mL	**4.90 * (0.19, 9.60)**	1.45 (−1.56, 4.47)	2.62 (−0.12, 5.35)
*p*-trend	**0.029 ***	0.159	**0.014 ***
**Waist Circumference Change**			
Decrease ≥480 mL	1.40 (−2.59, 5.38)	−1.40 (−4.07, 1.27)	1.30 (−2.74, 5.34)
Decrease 240 to <480 mL	1.60 (−2.17, 5.36)	−0.23 (−2.64, 2.18)	2.84 (−0.94, 6.63)
Decrease <240 mL	2.54 (−1.1, 6.18)	−0.12 (−2.53, 2.28)	2.62 (−1.45, 6.68)
No change to increase <240 mL	Reference	Reference	Reference
Increase 240 to <480 mL	3.94 (−0.87, 8.75)	0.61 (−2.46, 3.68)	**6.51 * (1.26, 11.76)**
Increase ≥480 mL	2.79 (−2.49, 8.08)	0.35 (−3.68, 4.38)	1.86 (−2.66, 6.39)
*p*-trend	0.073	0.434	0.109

Notes: * *p* < 0.05. Models were adjusted for sex, age, SES, physical activity, screen time, and age ∆ between visits.

## Data Availability

Not applicable.

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
