# Peer review of "Changes in Sugar Sweetened Beverage Intake Are Associated with Changes in Body Composition in Mexican Adolescents: Findings from the ELEMENT Cohort"

_nutrients, 2022, doi:10.3390/nu14030719_

Round 1

Reviewer 1 Report

Comments to the Authors of manuscript number: nutrients-1566428 entitled “Changes in Sugar Sweetened Beverage Intake is Associated with Changes in Body Composition in Mexican Adolescents: Findings from the ELEMENT Cohort”.

The authors have presented a study involved children from previous study involving their pregnant mothers. Now, the SSB use is followed during adolescence. It is very good study. Very rare such studies are presented where there is possibility to follow the life from the birth for a long time.

  1. L 35 – there also the abbreviation should be explained.
  2. Authors in very clear manner presented the problem of the SSB use in Mexico in general.
  3. L 77-85 – very well organized study, involved many time and effort
  4. L 113 – the producer of the set should be given
  5. L 122 – the moderate or vigorous physical activity should be defined
  6. L 206-214 – was this decrease noted a year or a period examined?

Reviewer 2 Report

Dear Authors

The scientific article “Changes in Sugar Sweetened Beverage Intake is Associated with Changes in Body Composition in Mexican Adolescents: Findings from the ELEMENT Cohort”, is interesting and well structured. Despite presenting a low originality because there are already several studies with the same objective, it is always important to demonstrate again, that the consumption of SSB is one of the causes associated with changes in body composition; especially in overweight and obese adolescents and that, this situation is observed in the Mexican adolescents evaluated in this study.

Accurate assessment of dietary exposure is challenging and hard and the association between the intake of a certain group of food and the change in body composition is a very difficult issue, laborious and not always with promising results. However, even with dietary limitations, this study gives important information to be taken into account in a country where the prevalence of overweight and obesity among adolescents ages 12 to 19 years increased to 38.4% in the most recent Mexican National Survey (ENSANUT 2018), as you mentioned.

Nevertheless, the paper needs some adjustments and some clarifications, especially at the level of the presentation of the results.

Here are my suggestions / corrections

Keywords

Line 32 – keywords should be different to the words of the title.

  1. Results

Table 2 and table 3 are the same.

The information on lines 176,177, 178 and 179 are nor based on current table 2 data, probably based in a different table 2. The same occurs for lines 189 to 193.

Please confirm and explain better.

Line 202 and 203 - Model 3: p=0.008 is not in accordance with the value in table 3.

Line 214 - Model 3: p=0.017 is not in accordance with the value in table 3.

Lines 217 to 220 – could explain better, maybe with values to be understandable.
